# Systematic Review of Nutrition Interventions to Improve Short Term Outcomes in Head and Neck Cancer Patients

**DOI:** 10.3390/cancers15030822

**Published:** 2023-01-29

**Authors:** Claire Leis, Anna E. Arthur, Xin Chen, Michael W. Greene, Andrew D. Frugé

**Affiliations:** 1Department of Nutritional Sciences, Auburn University, Auburn, AL 36849, USA; 2Department of Dietetics and Nutrition, University of Kansas Medical Center, Kansas City, KS 66160, USA; 3College of Nursing, Auburn University, Auburn, AL 36849, USA

**Keywords:** head and neck neoplasms, nutrition intervention, cancer control, nutritional status, quality of life

## Abstract

**Simple Summary:**

Head and neck cancer (HNC) patients are at increased risk of malnutrition prior to and during radiation, chemotherapy, and surgical treatment. Nutrition intervention is recommended by several organizations, but is not part of treatment guidelines. We conducted a systematic review of randomized controlled trials in HNC patients to determine whether nutrition interventions prevented weight loss, and improved quality of life, nutrient intake, and treatment tolerance. Interventions, including medical nutrition therapy provided by a registered dietitian and including oral nutrition supplements, had the most favorable outcomes in these patients.

**Abstract:**

Head and neck cancer (HNC) is associated with high rates of malnutrition. We conducted a systematic review and descriptive analysis to determine the effects of nutrition interventions on the nutrition status, quality of life (QOL), and treatment tolerance of HNC patients. PubMed, Web of Science, and Embase were searched to include all potentially relevant studies published between 2006–2022. Meta-analysis was not conducted due to heterogeneity of study designs and outcomes reported. Studies were categorized as nutrition interventions: (1) with oral nutrition supplements (ONS) and medical nutrition therapy (MNT) delivered by an RD; (2) with enteral nutrition (EN) support and MNT delivered by an RD; (3) with motivational interviewing and no ONS or EN; and (4) with ONS and no RD. Seven articles met inclusion criteria. Studies measured outcomes from immediately following treatment to 12 months post-treatment. Interventions resulted in benefits to lean mass/weight maintenance (three studies), QOL (two studies), nutrient intake adequacy (one study) and treatment tolerance (two studies). Nutrition counseling by a registered dietitian leads to improved nutrition status and QOL. Further research is needed to determine best practices related to timing of initiation, duration of nutrition intervention, as well as frequency of dietitian follow-up.

## 1. Introduction

Cancers of the head and neck, including oral cavity, pharynx, larynx, paranasal sinuses and nasal cavity, and salivary glands account for approximately 8% of all cancer diagnoses worldwide [1]. Despite accounting for a small number of cancer diagnoses, patients with head and neck cancer (HNC) are often the most malnourished among all cancer patients. Approximately 30% of HNC patients are malnourished prior to treatment initiation due to reduced food intake secondary to odynophagia, early satiety, fatigue, psychological, or other problems [2]. Primary treatment of HNC, including surgery and radiotherapy, sometimes with chemotherapy, exacerbates decreased food intake due to mucositis, xerostomia, dysphagia, or dysgeusia, leading to an additional 10% loss of pretherapy body weight loss. Weight loss of ≥20% during treatment is associated with increased risk of treatment interruption or cessation, infection, hospital readmission after treatment completion, and mortality [2]. Weight change as low as a 5% loss is associated with increased mortality risk [3].

The Academy of Nutrition and Dietetics evidenced-based nutrition practice guidelines for oncology patients recommends early intervention of medical nutrition therapy provided by a registered dietitian (RD) for cancer diagnoses of all types [4]. Furthermore, the recommendations suggest frequent monitoring of patients at high-risk for weight loss, including those with HNC [4]. The National Comprehensive Cancer Network (NCCN) provides specific recommendations for assessing and managing the nutrition status of individuals with HNC. It is recommended that weight be assessed regularly for significant weight loss (5% weight loss in one month, or 10% weight loss in six months) [5]. Signs of difficulty swallowing should also be assessed regularly, with consideration for abnormalities due to pain or tumor involvement [5]. The NCCN recommends patients with HNC receive continued follow-up with an RD until the patient has reached a post-treatment baseline [5].

Nutrition counseling is regarded as the first line of nutrition therapy. Nutrition counseling provides individualized nutrition recommendations that encourage the patient to maintain or increase calorie and protein intake utilizing foods from their usual diet. Oral nutrition supplements (ONS) are often recommended if oral intake is inadequate to meet the patient’s energy needs. Physical therapy is also recommended in order to promote anabolism, leading to the retention and utilization of nutrients. Drug therapy may also be utilized in severely malnourished patients to stimulate appetite and/or gut motility, decrease systematic inflammation, and/or hypercatabolism, or to increase muscle mass and improve anabolism [6,7]. 

It has been suggested, however, that the current recommendations are not fully integrated into practice. Studies have found oncology patients are often referred to an RD only after significant weight loss has occurred. A study by Lorton et al. found that 45% of patients referred to a RD should have been referred earlier [8]. Patients in the outpatient setting were significantly more likely (*p* = 0.042) to have received late nutrition intervention when compared to patients in the inpatient setting [8]. A study reported by Crowder et al. of dietary intervention preferences for post-treatment HNC survivors found that the majority of patients would have preferred nutrition intervention before treatment, and one-third of patients indicated the intervention should have occurred during or immediately after treatment [9]. 

The 2013 systematic review by Langius et al. reported findings from nutrition interventions in patients with HNC receiving radiation therapy (RT) or chemoradiation therapy (CRT) [10]. Results indicated individualized nutrition counseling to have beneficial effects on energy and protein intake, nutrition status, and QOL. Oral nutrition supplements were shown to have short-term effects on energy and protein intake, but inconsistent effects on nutrition status. Use of PEG tubes compared to nasogastric (NG) feeding showed beneficial effects short term, but no long-term effects on nutrition status. A comparison of prophylactic PEG feeding versus no prophylactic PEG feeding showed no beneficial effects on nutrition status and mortality and inconsistent effects on QOL [10]. The purpose of the current study was to conduct an updated systematic review and descriptive analysis of randomized controlled trials conducted for nutrition interventions in HNC patients including studies reporting outcomes on nutrition status, QOL, and treatment tolerance. 

## 2. Materials and Methods

### 2.1. Study Selection Critera

This systematic review was performed following the Preferred Reporting Items for Systematic Reviews and Meta-Analyses (PRISMA) statement. Using PubMed, Web of Science, and Embase databases, we conducted systematic searches to include all potentially relevant studies. Our search strategy included keywords for nutrition intervention (e.g., “diet therapy” or “nutrition status”) or nutrition counseling (e.g., “counsel“ or “counselled”), cancer (e.g., “cancerous” or “neoplasm”), head and neck cancer (e.g., “head and neck neoplasm”), and randomized controlled trial. The full search strategy is included in Appendix A. The database was searched for articles published from 2006—1 August 2022, which minimized potential overlap of studies included in the Langius et al. review [10]. 

### 2.2. Study Selection Criteria

Randomized controlled trials which met the following PICOS criteria (Table 1) were included within the systematic review: (1) patients with a diagnosis of HNC undergoing radiation alone or in combination with other treatments, (2) patients received nutrition intervention (interventions which offered nutrition counseling, ONS, enteral nutrition, immunonutrition, or a combination of these interventions were considered). Comparator nutrition of any type was included, such as maintenance of usual diet, usual care, or a placebo ONS), (3) studies addressed at least one of the following outcomes: nutrition status (e.g., PGSGA, weight loss), treatment tolerance, or quality of life, (4) studies with a follow-up time of 12 months post-treatment or less, (5) studies with quantitative outcomes. Only full-text, peer reviewed articles published in English were included in the systematic review. Studies were excluded for the following: (1) studies that reported malignancies other than HNC, (2) participants did not receive radiation treatment, (3) studies that did not include a nutrition intervention, (4) studies that included outcomes which did not measure nutrition status, treatment tolerance, or quality of life, and (5) studies that were ongoing or a follow-up time of greater than one year. 

### 2.3. Study Quality Assessment

Study quality was assessed by the following criteria: (1) Was the research question/hypothesis clearly stated? (2) Were the inclusion and exclusion criteria clearly stated? (3) Was a clear aim for the study stated? (4) Were the methods of the study clearly described? (5) Were the main findings (results) clearly described? (6) Were study limitations discussed? (7) Were outcomes (nutrition status, QOL, or treatment tolerance) identified using a validated measure? (8) Was a sample size justification via power analysis provided? (9) Are data analyses discussed? Two authors of the review, C. Leis and X. Chen, independently assessed each study using the criterion. Studies received either a 0, 1, or 2, depending on whether the criterion was unmet (0), partially met (1), or fully met (2). With nine total criteria, the total possible scores ranged between 0 and 18. Lower total scores indicated poorer study quality, and higher total scores indicated better quality. Scoring each study allowed researchers to gain a better understanding of the strength of study evidence. 

## 3. Results

### 3.1. Study Selection and Characteristics

The study selection is outlined in Figure 1. One hundred fifty-three articles were identified through database searching. Four articles were removed due to being duplicate records. One hundred forty-nine articles were screened by title and abstract. One hundred nine articles were excluded due to failure to meet inclusion criteria, including malignancies other than HNC, no nutrition intervention was provided, patients were not receiving chemotherapy or radiation therapy, the length of follow-up was greater than one year, studies lacking randomization, or initial review indicated different outcomes. Thirty-seven full-text articles were reviewed, and thirty studies were excluded due to having outcomes not matching the inclusion criteria, an absence of data, or not being quantitative. 

### 3.2. Study Quality Assessment

Appendix A reports the results of the study quality assessment, and the methodology used to analyze collected data is described. On average, studies included in the review scored 17 out of 18 and ranged between 15 and 18.

### 3.3. Risk of Bias Assessment

Risk of bias assessment is summarized in Figure 2. Included studies were identified as having either low risk (four studies) or some concern (three studies) of bias. Six RCTs adequately described their randomization process, however, one study was found to have some concern of bias due to providing no information about concealment of the allocation sequence [11,12]. One study received high concern of bias due to missing outcome data [13]. One study received some concern of bias [14], and two studies [11,15,16] received high risk of bias in the measurement of the outcomes due to outcome assessors being aware and possibly influencing the intervention received. One study was identified as having some concern of selection bias due to selecting results from multiple outcomes or analyses of the data [16,17]. 

### 3.4. Study Characteristics

Seven studies met the inclusion criteria for the review and are summarized in Table 2, with additional participant details of each study included in Appendix A. 

#### 3.4.1. RD Intervention with ONS

In this review, three studies included RD intervention with ONS. All ONSs are described in Appendix A. Patients who received nutrition counseling and two ONSs per day had less change in body weight (*p* = 0.006), increased protein-calorie intake (*p* < 0.001), and higher QOL scores (*p* < 0.01) at three months after RT than patients who received nutrition counseling alone [14]. In a similar study in which patients received nutrition counseling and only one ONS per day, patients were found to have no difference in body weight or QOL scores at three months after RT when compared to individuals who received nutrition counseling alone [13]. The final study involving RD intervention with ONS was performed in conjunction with progressive resistance training (PRT). Patients who received PRT, as well as an RD consultation and one ONS per day either during RT or after RT, had no significant difference in body weight (*p* = 0.818) between the two groups from baseline to week 14 [18]. 

#### 3.4.2. RD Intervention with Motivational Interviewing

Two studies focused on utilizing RD intervention with motivational interviewing. In the study by Britton et al., six hospitals providing RT to patients with HNC were randomized to a stepped-wedge trial, in which patients of the control group received standard treatment as defined by each hospital, before randomly progressing to the intervention group to receive motivational interviewing (MI) and cognitive behavioral therapy [16]. The dietitian-delivered intervention was based on four principles of behavior change, positing that behavior change would occur if the patient: (1) argues for the behavior themselves; (2) they devise the plan; (3) the plan is recorded externally; (4) believes it is important, achievable, and monitored. The goal was in increase the patient’s self-awareness of their motivation for change and sustainability of that after treatment. At home tools included a daily log of nutrition behaviors for accountability. Results indicated patients who received MI and cognitive behavioral therapy had significantly lower Patient-Generated Subjective Global Assessment (PG-SGA) scores (*p* = 0.03) and less percent weight loss (*p* = 0.03) than the control group [16]. Additionally, patients who received MI and cognitive behavioral therapy had significantly fewer interruptions in RT treatment (*p* = 0.04) and significantly better QOL (*p* < 0.01) [16]. Orell et al. compared intensive nutrition counseling (INC) to on-demand nutrition counseling (ODC) [19]. The experimental group, INC, received nutrition counseling at four different points during treatment: baseline, 2nd week of treatment, 4th week of treatment, and upon completion of chemoradiotherapy [19]. The control group, ODC, received nutrition counseling at baseline and as needed according to preset criteria including weight loss greater than 5% or symptoms significantly affecting oral intake [19]. Post-treatment follow-up indicated there to be no significant difference in nutrition status, including weight between the two groups [19]. Results indicated median PG-SGA scores to have significantly increased in both groups during treatment (*p* < 0.001) [19]. Patients of the INC group experienced weight loss at a greater rate than patients of the ODC group, 77% and 67%, respectively [19]. In regard to treatment tolerance, patients of the INC and ODC had similar rates of chemotherapy completion, 61% and 60%, respectively. Of the patients receiving radiotherapy, 91% of patients completed treatment as scheduled [19]. 

#### 3.4.3. Non-RD Intervention

Two studies included in the review provided interventions by a health professional other than an RD. One study compared a 12-week lifestyle intervention during treatment versus after treatment [11]. The multifactorial lifestyle intervention incorporated five components including: (1) physician referral and clinic support; (2) health education; (3) behavior change support; (4) an individualized exercise program; and (5) a group-based exercise program to include social support [11]. Results indicated there to be no significant difference in lean body mass (*p* = 0.756), BMI (*p* = 0.698), percent body fat (*p* = 0.741), or nutrition status (*p* = 0.846) between the two groups [11]. Additionally, no significant difference was seen in physical, functional, or anemia-specific QOL (*p* = 0.751) or for HNC-specific QOL (*p* = 0.503) [11]. The second study compared ONS in individuals receiving progressive resistance training (PRT); the experimental group received PRT and an ONS of creatinine and protein powder while the control group received PRT and an ONS of isocaloric placebo [20]. Results indicated both groups had decreased fat mass and increased lean body mass, though the treatment group had a significant increase in lean body mass (*p* < 0.001) during the 12-week intervention [20]. The intervention group also had a nonsignificant increase in body weight [20]. 

## 4. Discussion

This study systematically reviewed seven original articles related to the impact of nutrition intervention in HNC patients on nutrition status, QOL, and treatment tolerance. Extensive nutrition intervention using ONS, motivational interviewing, and progressive resistance training were all effective in minimizing weight loss and undesirable side-effects of treatment. 

It is established that oncology patients, including those with HNC, require proactive evaluation of nutrition status. However, as indicated in the previously mentioned study by Lorton et al., patients are often referred to an RD only after significant weight loss has occurred. A qualitative study of dietary counseling preferences in individuals with HNC by Crowder et al. indicated participants would have preferred to receive nutrition intervention prior to initiation of treatment [9]. 

There is, however, currently a lack of consensus regarding the duration of intervention throughout the treatment process. This updated literature review provides evidence from diverse study designs discussed below, which may provide greater insight into nutrition strategies to improve HNC outcomes affected by weight loss. 

The study by Ho et al. allowed patients to self-select nutrition intervention with a RD for either early, late, or no nutrition intervention during CCRT [21]. Patients of the early and late groups received a RD consultation at least every two weeks. Results indicated a significant change in body weight, early termination of chemotherapy, and incomplete planned radiotherapy rates among the early, late, and no nutrition counseling groups [21]. Such results emphasize the benefit of early, rather than late, nutrition intervention in regard to weight change and treatment tolerance. Furthermore, the results indicate nutrition counseling, regardless of timing, to have a positive effect on weight change and treatment tolerance when compared to no nutrition counseling [21]. 

The study by van den Berg et al. reported that HNC patients who received individualized nutrition counseling by an RD began gaining weight as early as two weeks after treatment [22]. Those who received nutrition counseling continued to gain weight through two months after treatment; patients who received standard treatment continued to lose weight [22]. Similarly, Ravasco et al. indicated patients who received nutrition counseling maintained their energy intake, maintained or improved QOL, and improved their nutrition status with a net average recovery of 4 kg at three months when compared to patients who received no nutrition counseling [23]. 

The observational study by Yang et al. of oncology patients with varying diagnoses indicated frequency of RD consultation during hospital admission to be associated with greater energy intake and less weight loss [24]. Each RD consultation was significantly associated with weight gain, which was continued through a follow-up duration of ≤six months [24]. Similarly, energy consumption increased with RD consultation frequency with significance seen upon consultation numbers three, four, five through seven, and eight and above [24]. Patients with cancers related to food ingestion, such as head and neck, were found to have clinically, but not statistically significant, improved energy intake, suggesting the beneficial effect of frequent RD consultations [24]. 

Further research is needed to establish best practices in regard to duration of nutrition intervention for patients with HNC. Immunotherapy is emerging as a promising treatment in advanced HNC, though limited data have been reported on how these treatments affect nutritional status. One recent study by Guller et al. reported decreased survival and efficacy of anti-PD-1 or anti-CTLA-4 treatments in patients with poor nutrition status and/or a trend in pre-treatment weight loss [25]. In a study of several cancer types employing these same therapies, Johannet et al. reported similar prognostic results [26]. Finally, one study of pembrolizumab in recurrent/metastatic HNC patients reported pre-treatment BMI as an independent predictor of overall and progression-free survival times [27]. These and previously discussed results indicate that early, rather than late or no nutrition intervention, resulted in better prognosis, as well as less change in body weight, less early termination of chemotherapy, and decreased rates of incomplete planned radiotherapy [21]. Additionally, a study of dietary intervention preferences for post-treatment HNC survivors indicated that the majority of patients would have preferred nutrition intervention before, during, or immediately after treatment [9]. 

It is necessary to note that the seven studies included in this review were conducted in seven different countries. This could be considered a strength of the review, given that diverse cultures and practices for improving nutrition outcomes in patients with HNC were included. However, there is likely heterogeneity in the nutritional care provided within each culture, including adherence to nutrition guidelines, which may have had an impact on the findings of this review. 

A major limitation of this review was the heterogeneity of interventions (e.g., utilization of ONS or RD, and timing relative to treatment) between studies. Another confounding and problematic outcome is whether weight loss may be beneficial in individuals with high BMI. In fact, one recent study observed greater overall survival when patients with greater pre-treatment body mass lost >5% body weight [28]. Another major confounder in overall survival is post-treatment diet. Several epidemiological studies have observed survival benefits to patients consuming more servings of vegetables [29], and whole, minimally processed foods [30]. 

While most included studies followed participants through three months, additional research is needed to assess possible benefits of longer follow-up. This systematic review included 1025 patients with HNC who received nutrition interventions including: ONS and MNT delivered by an RD, motivational interviewing, and ONS without an RD. Results indicated that when compared to control groups which received general routine care with nutrition intervention as needed, interventions resulted in benefits to lean mass/body weight maintenance (three studies), QOL (two studies), nutrient intake adequacy (one study) and treatment tolerance (two studies).

## 5. Conclusions

Randomized controlled trials utilizing nutrition interventions to improve outcomes in head and neck cancer patients undergoing treatment have differed in intervention type, duration, geographic location, and measures. Interventions that include medical nutrition therapy provided by registered dietitians and provision of oral nutrition supplements are most promising in improving nutrition status, quality of life, and treatment tolerance, as well as preventing lean muscle loss. Future studies should focus on the timing, intensity, and duration of nutrition intervention initiation related to treatment in order to decrease morbidity and mortality in this population. 

## Figures and Tables

**Figure 1 cancers-15-00822-f001:**
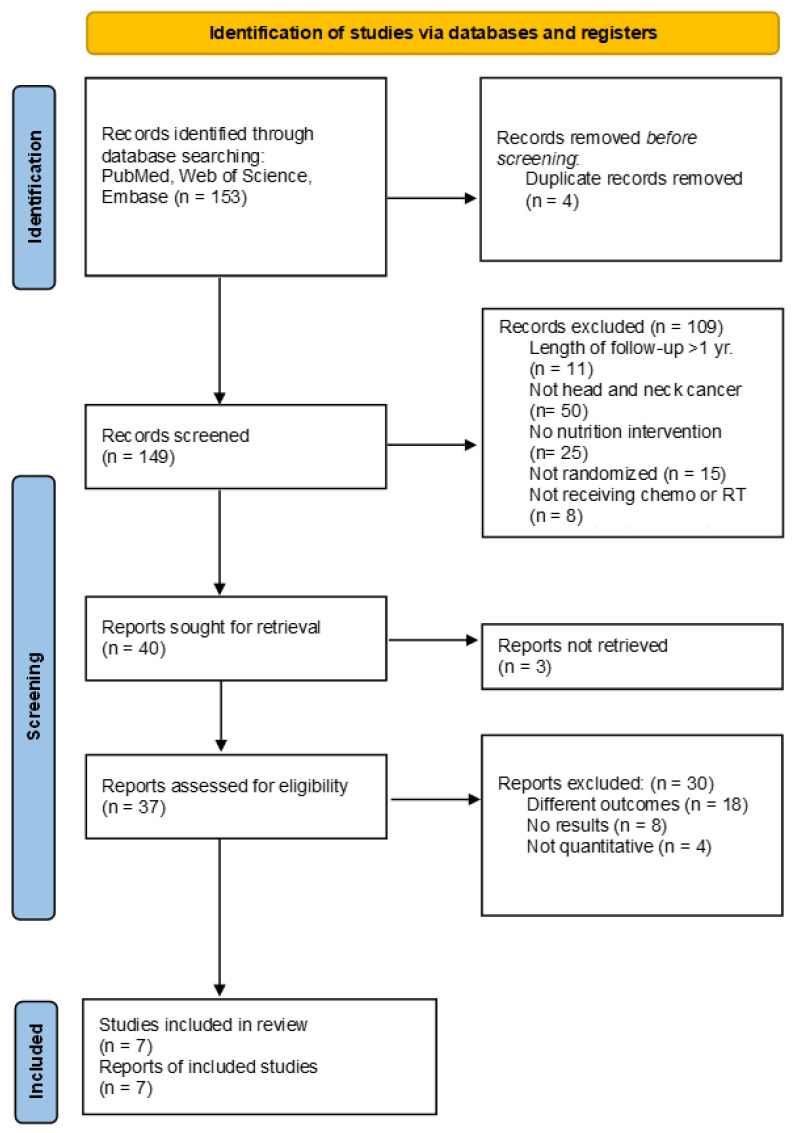
PRISMA flow diagram of studies reporting nutrition interventions in head and neck cancer patients.

**Figure 2 cancers-15-00822-f002:**
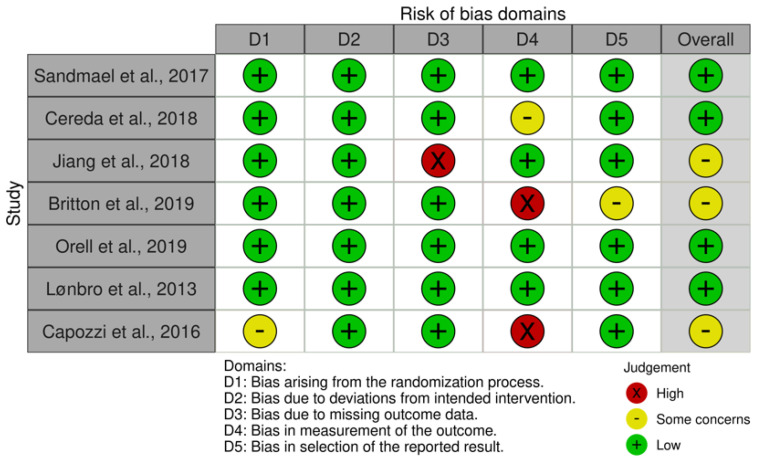
Risk of Bias Assessment [11,13,14,16,18,19,20].

**Table 1 cancers-15-00822-t001:** PICOS Table.

Acronym:	Definition:	Description:
P	Patient or problem	Human subjects with a head and neck cancer diagnosis
I	Intervention	(1) Oral nutrition supplements (ONS) and MNT by a registered dietitian (RD), (2) enteral nutrition and MNT by an RD, (3) motivational interviewing by an RD, and (4) ONS and no RD.
C	Comparison	Not applicable
O	Outcomes	How nutrition intervention is associated with nutrition status, QOL, and treatment tolerance
S	Study Design	Quantitative studies

**Table 2 cancers-15-00822-t002:** Characteristics of studies reporting nutrition interventions in patients with head and neck cancers.

Author, Year	Intervention	Main Inclusion Criteria	Sample Size	Length of Follow-Up	Nutrition Status	Treatment Tolerance	Quality of Life	Country
	**RD intervention with ONS**							
Sandmael et al., 2017 [18]	Intervention: PRT and nutrition intervention of RD consult and 1 supplement/day during RT; Control: Post-treatment intervention	Diagnosis of HNSCC with referral for curative RT with or without chemo	41	Two months	No difference between the two groups from baseline to week 14 for change body weight (*p* = 0.818) or change in muscle mass (*p* = 0.821)			Norway
Cereda et al., 2018 [14]	Intervention: Nutrition counseling and two ONS per day; Control: Nutrition Counseling	Newly diagnosed HNC patients suitable for RT or RT plus systematic treatment	159	Three months after the end of RT	Counseling plus ONS resulted in less change in body weight (*p* = 0.006), increased protein-calorie intake (*p* < 0.001), and increased protein intake (*p* < 0.001) than nutritional counseling alone	No difference in tolerance to anti-cancer treatments were in the two groups, however, patients receiving ONS were less likely to require RT and/or ST dose reduction or complete suspension	Counseling plus ONS resulted in higher QOL scores (*p* < 0.001)	Italy
Jiang et al., 2018 [13]	Intervention: ONS once daily; Control: No extra nutritional supplements were provided	HNC (NPC), stage 3 or 4 receiving chemoradiation	100	Three months after the end of CRT	ONS group had higher body weight (*p* = 0.036) and BMI (*p* = 0.021) at the end of CRT; No difference between groups at three months post-CRT in weight (*p* = 0.71), BMI (*p* = 0.608), FFM (*p* = 0.809), FFMI (*p* = 0.800)		Patients in the ONS group had higher QOL at the end of CRT (*p* = 0.045); No between group difference at three months post-CRT in QOL (*p* = 0.294)	China
	**RD intervention with Motivational Interviewing (MI)**							
Britton et al., 2019 [16]	Intervention: RD provides MI and cognitive behavioral therapy (CBT); Control: RD provides treatment as usual	HNC requiring RT or concurrent chemoradiation with curative intent	307	12 weeks after the end of RT	Patients who received MI and CBT had significantly better (lower) PG-SGA scores (*p* = 0.03) and less percent weight loss (*p* = 0.03) than control	Patients who received MI and cognitive behavioral therapy had significantly less interruptions in RT treatment (*p* = 0.04).	Patients who received MI and cognitive behavioral therapy had significantly better QOL (*p* < 0.01)	Australia
Orell et al., 2019 [19]	Intervention: RD provides intensive nutritional counseling (INC); Control: RD provides on-demand nutrition counseling (ODC)	Locally advanced HNC receiving curative treatment with combined surgery and adjuvant (chemo) radiotherapy, or definitive (chemo) radiotherapy	65	Primary outcomes assessed at end of treatment; Survival measured—median of 43 months	No difference in nutrition measures between groups. PG-SGA scores increased for all patients during treatment (*p* < 0.001), 77% of patients had critical weight loss in the INC vs. 67% in the ODC group. Weight loss was greater in the group with baseline OW vs. normal BMI (*p* < 0.001).			Finland
	**Non-RD intervention**							
Lønbro et al., 2013 [20]	Intervention: PRT + ONS consisting of creatine and protein powder; Control: PRT + isocaloric placebo (maltodextrin)	HNC, terminated curative RT treatment +/− chemotherapy	30	12 weeks	Both groups had increased LBM, intervention group had LBM increase (*p* < 0.0001), but not BW.			Denmark
Capozzi et al., 2016 [11]	Intervention: 12-week immediate lifestyle intervention (ILI); Control: 12-week delayed intervention (DLI)	HNC scheduled to receive radiation or concurrent chemoradiation treatment	60	12 months	No difference between the two groups across the 24 weeks for lean body mass (*p* = 0.756), BMI (*p* = 0.698), percent body fat (*p* = 0.741), or nutrition status (*p* = 0.846)		No difference between the two groups across the 24 weeks for physical, functional, or anemia-specific QOL (*p* = 0.751) or for HNC-specific QOL (*p* = 0.503)	Canada

PRT: progressive resistance training; RD: registered dietitian; RT: radiation therapy; HNSCC: head and neck squamous cell carcinoma; ONS: oral nutrition supplements; HNC: head and neck cancer; QOL: quality of life; CRT: chemo radiation therapy; FFM: fat free mass; FFMI: fat-free mass index; MI: motivational interviewing; PG-SGA: patient-generated subjective global assessment; OW: overweight; BMI: body mass index.

## Data Availability

Not applicable.

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
