# Peer review of "Systematic Review of Nutrition Interventions to Improve Short Term Outcomes in Head and Neck Cancer Patients"

_cancers, 2023, doi:10.3390/cancers15030822_

Round 1

Reviewer 1 Report

Please comment that the study is a descriptive rather quantitative analysis (you rightly commented that a meta analysis was not performed due to the heterogeneity of the studies)

Author Response

Please comment that the study is a descriptive rather quantitative analysis (you rightly commented that a meta analysis was not performed due to the heterogeneity of the studies)

Thank you for reviewing our manuscript and providing additional guidance. We have added verbiage to clarify the descriptive nature of this study as noted below.

The second sentence of the abstract now reads: “We conducted a systematic review and descriptive analysis to determine the effects of nutrition interventions on the nutrition status, quality of life (QOL), and treatment tolerance of HNC patients.”

Beginning on Line 87: “The purpose of the current study was to conduct an updated systematic review and descriptive analysis of randomized controlled trials conducted in the past 15 years for nutrition interventions in HNC patients including studies reporting outcomes on nutrition status, QOL, and treatment tolerance.”

Reviewer 2 Report

This is a modern (studies from 2006 - 2022) systematic review of nutritional interventions studied in head and neck cancer patients before, during or immediately after definitive therapy that had to include radiation therapy. 

There is few consensus on nutritional guidelines for cancer patients but this study  helps to acutely define meaningful interventions in a patient population that is typically malnourished at time of presentation. I do have some thoughts/recommendations:

Methods:

- Why was 2006 chosen as the oldest study year?

- Study excluded trials greater than 1 year follow up, why? If these excluded studies looked at interventions given during treatment, it could be beneficial to include. Authors themselves noted that how long nutritional interventions should be applied is vague, these excluded studies can help answer that questions.

Results:

- Supplemental Table 2 - said 2 authors did the study quality assessment but I only see one score for each study. I do see the Mean and SD, but adding both scores would be complete

- Section 3.3.1 - describing what the specific ONS were in these studies would add to the manuscript

- Section 3.3.2 - Briefly define what motivational interviewing is and what did it entail in that study

- Discussion: please speak more about the weakness of the review - how a metanalysis could not be performed due to the very nature of nutritional studies (no clear guidance on what is an important outcome to measure - BMI/Weight?, decreased RT disruptions? survival?). Also, a brief few sentences on (though its beyond the scope of the manuscript) how important nutrition is for long term cancer survivors.

Author Response

Reviewer 2:
This is a modern (studies from 2006 - 2022) systematic review of nutritional interventions studied in head and neck cancer patients before, during or immediately after definitive therapy that had to include radiation therapy.
There is few consensus on nutritional guidelines for cancer patients but this study helps to acutely define meaningful interventions in a patient population that is typically malnourished at time of presentation. I do have some thoughts/recommendations:

We sincerely appreciate your thoughtful review and feedback. We have incorporated your recommendations as described below, all of which have improved the quality of our manuscript.

Methods:

- Why was 2006 chosen as the oldest study year?

The relevant studies from the Langius et al. review were from 1985-2005, so we chose to provide an updated/current review.

- Study excluded trials greater than 1 year follow up, why? If these excluded studies looked at interventions given during treatment, it could be beneficial to include. Authors themselves noted that how long nutritional interventions should be applied is vague, these excluded studies can help answer that questions.

This is an excellent point. Studies with longer follow-up have only reported survival outcomes, since assessing QOL and nutritional status require more intensive efforts than retrospective chart reviews or an audit of death certificates. Regarding duration of nutrition interventions, observational studies indicate that interventions that are early (Ho et al., 2021) and more frequent (Yang et al., 2018) are the strongest predictors of benefits to QOL and nutritional status. Reasons for this would be the real time feedback and troubleshooting to overcome intake barriers associated with treatment toxicity.

Results:

- Supplemental Table 2 - said 2 authors did the study quality assessment but I only see one score for each study. I do see the Mean and SD, but adding both scores would be complete

Supplementary Table 2 now lists all scores including third reviewer’s score to resolve discrepancies.

- Section 3.3.1 - describing what the specific ONS were in these studies would add to the manuscript

Supplementary Table 3 lists product name, manufacturer, and calories/protein for each of the ONS.

- Section 3.3.2 - Briefly define what motivational interviewing is and what did it entail in that study

The following has been added beginning on line 190: “The dietitian-delivered intervention was based on four principles of behavior change, pos-iting that behavior change would occur if the patient 1) argues for the behavior them-selves; 2) they devise the plan; 3) the plan is recorded externally; 4) believes it is important, achievable and monitored. The goal was in increase the patient’s self-awareness of their motivation for change and sustainability of that after treatment. At home tools included a daily log of nutrition behaviors for accountability.”

- Discussion: please speak more about the weakness of the review - how a metanalysis could not be performed due to the very nature of nutritional studies (no clear guidance on what is an important outcome to measure - BMI/Weight?, decreased RT disruptions? survival?). Also, a brief few sentences on (though its beyond the scope of the manuscript) how important nutrition is for long term cancer survivors.

Thank you for this guidance. The following has been added beginning on Line 294: “A major limitation of this review was the heterogeneity of interventions (e.g., utilization of ONS or RD, and timing relative to treatment) between studies. Another confounding and problematic outcome is whether weight loss may be beneficial in individuals with high BMI. In fact, one recent study observed greater overall survival when patients with greater pre-treatment body mass lost > 5% body weight. Another major confounder in overall survival is post-treatment diet. Several epidemiological studies have observed survival benefits to patients consuming more servings of vegetables, and whole, minimally processed foods.”